

# Prognostic nutritional index (PNI) as an influencing factor for in-hospital mortality in patients with stroke-associated pneumonia: a retrospective study

Ke Xie*, Chuan Zhang*, Shiyu Nie, Shengnan Kang, Zhong Wang and Xuehe Zhang

Department of Intensive Care Unit, The Third People's Hospital of Chengdu, Chengdu, China
* These authors contributed equally to this work.

## ABSTRACT

**Background:** Stroke-associated pneumonia (SAP) significantly increases patients' risk of death after stroke. The identification of patients at high risk for SAP remains difficult. Nutritional assessment is valuable for risk identification in stroke patients. The aim of this study was to evaluate the relationship between prognostic nutritional index (PNI) levels and in-hospital mortality in SAP patients.

**Methods:** A total of 336 SAP patients who visited the Third People's Hospital of Chengdu from January 2019 to December 2023 were included in this study, and PNI were calculated based on the results of admission examinations. Linear regression was used to analyze the influencing factors of baseline PNI in SAP patients. Logistic regression as well as restricted cubic splines (RCS) were used to analyze the relationship between baseline PNI levels and hospital mortality events in SAP patients. Receiver operating characteristic (ROC) curves were plotted to assess the predictive value of PNI for in-hospital mortality by area under the curve (AUC).

**Results:** Thirty out of 336 SAP patients presented with in-hospital mortality and these patients had significantly lower PNI levels. In our study, PNI levels were influenced by age, body mass index, and total cholesterol. Increased PNI levels are an independent protective factor for the risk of in-hospital mortality in SAP patients (OR: 0.232, 95% CI [0.096–0.561], $P = 0.001$). There was a nonlinear correlation between PNI and in-hospital mortality events ($P$ for nonlinear <0.001). In terms of predictive effect, PNI levels were more efficacious in predicting in-hospital mortality in SAP patients with higher sensitivity and/or specificity compared to individual indicators (AUC = 0.750, 95% CI [0.641–0.860], $P < 0.001$).

**Conclusion:** PNI levels in SAP patients were associated with the short-term prognosis of patients, and SAP patients with elevated PNI levels had a reduced risk of in-hospital mortality.

Corresponding author
Xuehe Zhang, zxhwydh@163.com

## INTRODUCTION

Stroke as an acute cerebrovascular disease is a common cause of adult disability (*Katan & Luft, 2018*), and the cumulative incidence of stroke has been increasing yearly with the

increase of various risk factors (*Avan et al., 2019*). Acute ischemic stroke (AIS) accounts for a large proportion of stroke patients and is a common type of stroke (*Truelsen et al., 2003*). Stroke induces a localized inflammatory immune response leading to immune suppression and stroke-associated infections (*Haeusler et al., 2008*; *Liu et al., 2018*). Various complications leading to poor prognosis occur frequently during hospitalization for stroke, among which stroke-associated pneumonia (SAP) is common. Patients at high risk for SAP often face immunosuppression and dysphagia from stroke (*Teramoto, 2009*; *Hoffmann et al., 2017*). In addition to higher morbidity, SAP leads to serious consequences, such as prolonged hospitalization, poor functional regression, and increased morbidity and mortality (*Wilson, 2012*; *Hannawi et al., 2013*). Although some therapeutic and preventive measures have been taken for this common complication leading to death in stroke patients, the improvement in adverse outcomes has not been significant. A number of other clinical characteristics have also been suggested to be associated with the risk of SAP, such as advanced age, male gender, and chronic obstructive pulmonary disease (COPD) (*Gong et al., 2016*; *Hoffmann et al., 2017*). There are also studies that have developed clinical scoring systems to predict the occurrence of SAP in stroke patients, but this may take additional time or require more clinical information about the patients. Although there is some understanding of SAP, improving the prognosis of patients with SAP remains a challenge, as the prognosis of patients with SAP is influenced by a variety of factors, and the risk of death in patients with SAP has not yet been adequately identified in clinical practice. Malnutrition can lead to decreased immunity and infections (*Keusch, 2003*). Stroke patients are often malnourished, and stroke-induced impaired mobility and swallowing dysfunction can lead to malnutrition and affect patients' quality of life (*Sabbouh & Torbey, 2018*). Previous studies in patients with AIS have shown that malnutrition is an important predictor of poor clinical prognosis (*Yoo et al., 2008*). Stroke complications and multiple risk factors increase the risk of malnutrition, and in turn malnutrition is involved in the pathogenesis of neurologic disorders affecting the functional prognosis of patients (*Burgos et al., 2018*). Effective nutritional interventions to improve nutritional status can reduce infectious complications.

The prognostic nutritional index (PNI) is a score that can be calculated from laboratory tests to reflect a patient's immunonutritional status (*Buzby, 1979*). The clinical predictive value of PNI has been demonstrated in established studies, especially in cardiovascular, respiratory and oncology patients (*Wang et al., 2018*; *Wang, Zhao & He, 2021*; *De Rose et al., 2024*). In young AIS patients, PNI levels have a predictive value for the patient's 90-day prognosis (*Wang et al., 2023*). It has also been shown that PNI levels are associated with the risk of developing infections during hospitalization in patients with AIS (*Nergiz & Ozturk, 2023*). To our knowledge, the relationship between PNI levels and short-term prognosis in hospitalized stroke patients with comorbid pneumonia complications has not been clarified. Our study aimed to clarify the relationship between baseline PNI levels and the risk of death during hospitalization in SAP patients.

## METHODS

### Study population

This was a retrospective study that consecutively included patients with AIS who attended The Third People's Hospital of Chengdu from January 2019 to December 2023 and were diagnosed with AIS based on the results of cranial computed tomography or cranial magnetic resonance imaging within 24 h. Patients with AIS who developed lower respiratory tract infections within 7 days of stroke onset were included (*Smith et al., 2015*). Patients with traumatic brain injury, transient ischemic attack, subarachnoid hemorrhage, acute or chronic infection, chronic hepatic and renal insufficiency, autoimmune disease, hematological disorders, cancer and incomplete clinical information were excluded. Finally, 336 patients with SAP were included in our study. This study retrospectively collected and statistically analyzed patients' prior clinical information, and patients' health and empowerment were not compromised. Patients participating in this study were informed about the information, and patients or patients' families did not refuse the use of their medical records, so signed informed consent was waived for this study. This study was approved by the Medical Ethics Review Committee of The Third People's Hospital of Chengdu (2024-S-245).

### Data collection and measures

Demographic and clinical data, including age, sex, hypertension status, type 2 diabetes mellitus (T2DM), COPD, atrial fibrillation (AF), smoking history, height, weight, mechanical ventilation (MV) status, blood pressure and National Institutes of Health Stroke Scale (NIHSS) were recorded by trained specialized personnel. A blood sample was collected from patients within 24 h of admission, and the hospital's testing center performed the testing. The PNI was calculated on the basis of the patient's admission albumin (ALB) and lymphocyte (LYM) *via* the formula $5 \times \text{LYM} (10^9/\text{L}) + \text{serum ALB (g/L)}$. Hypertension was defined as either a blood pressure measurement of $\geq 140/90$ mmHg on separate days or the use of antihypertensive medication. T2DM was defined as a random blood glucose concentration greater than 11.1 mmol/L, a fasting blood glucose concentration $\geq 7.0$ mmol/L or current use of hypoglycemic medication. Smoking was defined as smoking at least 10 cigarettes a day for a period of at least a year. BMI over 25 kg/m$^2$ is considered overweight or obese.

### Statistical analyses

SPSS 25.0 (IBM, Armonk, NY, USA) and R 4.2.1 were used for data analysis. Measurement information was tested for normality. Measurement information was expressed as mean ± standard deviation or median (interquartile spacing) by normality test, and comparisons between two groups were made by t-test or rank-sum test. Count data were expressed as the number of cases (percentage), and comparisons between two groups were made using the chi-square test. Linear regression analysis was used to assess the factors influencing the PNI in SAP patients. Univariate and multivariate logistic regression analyses were used to assess the relationship between the PNI and the risk of experiencing in-hospital mortality in SAP patients. The nonlinear relationship between PNI levels and in-hospital mortality

**Table 1 Baseline characteristics of patients.**

| | Non in-hospital mortality ($n = 306$) | In-hospital mortality ($n = 30$) | $P$ |
|---|---|---|---|
| Age, years | 61 ± 13 | 67 ± 12 | 0.012 |
| Male, $n$ (%) | 232 (75.8) | 18 (60.0) | 0.058 |
| Hypertension, $n$ (%) | 181 (59.2) | 15 (50.0) | 0.332 |
| T2DM, $n$ (%) | 105 (34.3) | 14 (46.7) | 0.177 |
| COPD, $n$ (%) | 97 (31.7) | 17 (56.7) | 0.006 |
| AF, $n$ (%) | 55 (18.0) | 11 (36.7) | 0.014 |
| Smoking, $n$ (%) | 127 (41.5) | 7 (23.3) | 0.052 |
| BMI, kg/m$^2$ | 25.62 ± 3.52 | 23.75 ± 2.12 | 0.001 |
| NIHSS | 6 (4–8) | 6 (4–10) | 0.055 |
| MV, $n$ (%) | 24 (7.8) | 5 (16.7) | 0.101 |
| SBP, mmHg | 129 ± 23 | 120 ± 23 | 0.102 |
| DBP, mmHg | 77 ± 14 | 69 ± 14 | 0.028 |
| LYM ($10^9$/L) | 1.64 (1.12–2.49) | 1.00 (0.53–1.69) | <0.001 |
| ALB (g/L) | 39.84 (36.89–42.10) | 35.57 (29.98–37.93) | <0.001 |
| RBG (mmol/L) | 7.23 (5.53–10.34) | 10.31 (7.08–15.86) | 0.001 |
| TG (mmol/L) | 1.54 (0.98–2.41) | 1.19 (0.82–1.81) | 0.115 |
| TC (mmol/L) | 4.00 (3.25–4.74) | 3.80 (3.12–4.12) | 0.130 |
| HDL-C (mmol/L) | 0.91 (0.78–1.10) | 1.00 (0.78–1.20) | 0.088 |
| LDL-C (mmol/L) | 2.44 (1.80–3.22) | 2.30 (1.98–2.83) | 0.479 |
| PNI | 48.32 (43.86–54.15) | 40.60 (35.26–47.09) | <0.001 |

**Note:**
Abbreviations: T2DM, type2 diabetes mellitus; COPD, chronic obstructive pulmonary disease; AF, atrial fibrillation; BMI, body mass index; NIHSS, National Institutes of Health Stroke Scale; MV, Mechanical ventilation; SBP, systolic blood pressure; DBP, diastolic blood pressure; LYM, lymphocyte count; ALB, albumin; RBG, random blood glucose; TG, triglyceride; TC, total cholesterol; HDL-C, high-density lipoprotein cholesterol; LDL-C, low-density lipoprotein cholesterol; PNI, prognostic nutritional index.

in SAP patients was assessed by restricted cubic spline (RCS). Receiver operating characteristics (ROC) curve and area under the curve (AUC) were used to assess the predictive value of PNI levels for patient death. A $P$ value <0.05 was considered to indicate statistical significance.

## RESULTS

The mean age of the 336 SAP patients was 61 ± 13 years, with 250 males (74.4%) and 86 females (25.6%), and in-hospital mortality occurred in 30 SAP patients. There was a statistically significant difference in Age, COPD, AF, BMI, DBP, LYM, ALB, and RBG among SAP patients with in-hospital and non-in-hospital mortality ($P < 0.05$). PNI levels were significantly lower in SAP patients who died in-hospital compared to the non-in-hospital group ($P < 0.001$) (Table 1).

To further clarify the factors influencing the PNI in SAP patients, one-way linear regression analysis was conducted and revealed that age, BMI, and RBG, TG, and TC concentrations were associated with the PNI. Furthermore, the above indicators were included in multifactorial linear regression analysis, which revealed that age, BMI and TC concentrations influenced the PNI in SAP patients (Table 2).

**Table 2 Linear regression analysis of PNI levels and influencing factors in SAP patients.**

| | Univariable | | | Multivariable | | |
|---|---|---|---|---|---|---|
| | β | Standard β | P | β | Standard β | P |
| Age | −0.199 | −0.329 | <0.001 | −0.152 | −0.252 | <0.001 |
| BMI | 0.525 | 0.229 | <0.001 | 0.273 | 0.119 | 0.025 |
| RBG | −0.216 | −0.132 | 0.015 | −0.111 | −0.068 | 0.182 |
| TG | 1.582 | 0.245 | <0.001 | 0.691 | 0.107 | 0.051 |
| TC | 1.703 | 0.239 | <0.001 | 1.265 | 0.177 | 0.001 |

Note:
Abbreviations: BMI, body mass index; RBG, random blood glucose; TG, triglyceride; TC, total cholesterol; T2DM, type2 diabetes mellitus.

**Table 3 Logistic regression analysis of in-hospital mortality in SAP patients.**

| | Univariable OR (95% CI) | P | Multivariable OR (95% CI) | P |
|---|---|---|---|---|
| Male | 0.478 [0.220–1.039] | 0.063 | | |
| Age | 1.041 [1.010–1.073] | 0.010 | 0.998 [0.962–1.036] | 0.910 |
| BMI ≥ 25 | 0.173 [0.065–0.464] | <0.001 | 0.190 [0.066–0.544] | 0.002 |
| NIHSS | 1.052 [0.998–1.109] | 0.059 | | |
| RBG | 1.103 [1.037–1.171] | 0.002 | 1.073 [1.002–1.149] | 0.044 |
| Smoking | 0.429 [0.179–1.030] | 0.058 | | |
| T2DM | 1.675 [0.787–3.564] | 0.181 | | |
| Hypertension | 0.691 [0.326–1.464] | 0.334 | | |
| AF | 2.642 [1.190–5.867] | 0.017 | 1.144 [0.446–2.932] | 0.780 |
| COPD | 2.818 [1.316–6.032] | 0.008 | 2.385 [1.009–5.638] | 0.048 |
| MV | 2.350 [0.825–6.694] | 0.110 | | |
| PNI ≥ 45 | 0.187 [0.083–0.424] | <0.001 | 0.232 [0.096–0.561] | 0.001 |

Note:
Abbreviations: BMI, body mass index; NIHSS, National Institutes of Health Stroke Scale; RBG, random blood glucose; T2DM, type2 diabetes mellitus; AF, atrial fibrillation; COPD, chronic obstructive pulmonary disease; MV, Mechanical ventilation; PNI, prognostic nutritional index.

Based on previous studies, PNI ≥ 45 can be considered not to be a nutritional risk (*Dursun et al., 2022*; *Matsuo, Fujita & Amagai, 2022*). With reference to the comparative results of baseline information and clinical practice, regression analyses incorporated demographic information such as gender and age, as well as variables such as T2DM, hypertension, AF, and COPD. The results showed that age, BMI ≥ 25, RBG, AF, COPD, and PNI ≥ 45 were influential factors for in-hospital mortality in SAP patients, and the above variables were further included in the multifactorial regression analysis, which showed that BMI ≥ 25 (OR: 0.190, 95% CI [0.066–0.544], $P$ = 0.002), RBG (OR: 1.073, 95% CI [1.002–1.149], $P$ = 0.044), COPD (OR: 2.385, 95% CI [1.009–5.638], $P$ = 0.048), PNI ≥ 45 (OR: 0.232, 95% CI [0.096–0.561], $P$ = 0.001) were independent influences on in-hospital mortality in SAP patients (Table 3). The relationship between baseline PNI and in-hospital mortality events in SAP patients is shown in Fig. 1, and the results of the RCS

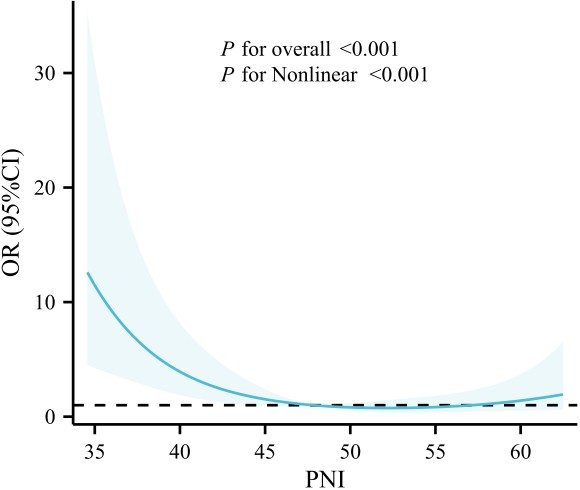

**Figure 1** Restricted cubic spline analysis of the relationship between PNI and in-hospital mortality. Statistical adjustments were made for age, sex, hypertension, T2DM, smoking, and BMI.

**Table 4 Predictive value of different indexes for in-hospital mortality in SAP patients.**

|  | AUC (95% CI) | Sensitivity (%) | Specificity (%) | P |
|---|---|---|---|---|
| LYM | 0.708 [0.595–0.821] | 66.7 | 64.7 | <0.001 |
| ALB | 0.746 [0.647–0.845] | 70.0 | 68.0 | <0.001 |
| PNI | 0.750 [0.641–0.860] | 73.3 | 68.0 | <0.001 |

**Note:**
Abbreviations: LYM, lymphocyte count; ALB, albumin; PNI, prognostic nutritional index.

curve analysis showed a nonlinear relationship between PNI and adverse endpoint events after adjusting for confounders such as age, sex, hypertension, T2DM, smoking, and BMI.

We further compared LYM, ALB, and PNI calculated by combining the above metrics for their predictive value of in-hospital mortality in patients with SAP. The results revealed that the AUC for LYM was 0.708 (95% CI [0.595–0.821], $P < 0.001$), the AUC for ALB was 0.746 (95% CI [0.647–0.845], $P < 0.001$), and the AUC for the PNI was 0.750 (95% CI [0.641–0.860], $P < 0.001$). PNI has a greater predictive value than any single indicator, and it also possesses a higher sensitivity and/or specificity (Table 4).

## DISCUSSION

Our study found that baseline PNI levels at admission in SAP patients were significantly lower in the in-hospital mortality group. Further analysis showed that PNI levels in SAP patients were associated with age, BMI, and TC levels. Higher PNI levels play an independent protective role for the short-term prognosis of patients.

With the prevalence of stroke-related risk factors increasing every year, the number of deaths due to stroke is also increasing dramatically in China. Data from the China Stroke Surveillance Report indicate that the hospitalized mortality rate for stroke patients will be 1.4% in 2020 (*Tu, Wang & Special Writing Group of China Stroke Surveillance Report, 2023*). The outcome of stroke patients is influenced by a number of factors, and studies on

post-stroke morbidity and mortality have shown that pneumonia is associated with morbidity and mortality in stroke patients (*Saposnik et al., 2008*). Complicated infections are very common after stroke, and factors including immune dysregulation, impaired consciousness, and aspiration all increase a patient's susceptibility to pulmonary infection (*Engel et al., 2007*). A variety of complications and neurologic injuries result in a generally poorer prognosis for these patients. Even with some preventive treatment and care process optimization, the incidence of SAP in the early post-stroke period remains high and significantly increases the risk of adverse outcomes. A large cross-sectional study in 2018 showed that the main comorbidity in Chinese stroke patients was pneumonia or lung infection, of which 10.1% were ischemic stroke (IS) patients and 31.4% were intracerebral hemorrhage (ICH) stroke patients (*Wang et al., 2022*). Although the proportion of ICH patients with pulmonary infection comorbidities was high among them, data from Hospital Quality Monitoring System and Bigdata Observatory Platform for Stroke of China showed that more than 80% of stroke patients admitted in 2020 were IS (*Wang et al., 2022*). Considering that IS is more common among stroke patients, AIS patients were selected as the study population in this study.

The identification of risk factors for SAP remains problematic, for example, the assessment of dysphagia still lacks recognized standards (*Hoffmann et al., 2017*). Although previous studies have attempted prophylactic antibiotic therapy, the improvement in the risk of death is not significant and promotes antibiotic resistance in bacteria (*Grau, Urbanek & Palm, 2010*; *Kishore et al., 2018*). Inflammatory responses are involved in the regulation of brain damage and peripheral organ dysfunction after stroke. Immunosuppression occurring after stroke worsens neurologic prognosis while increasing the risk of infection (*Howard & Simmons, 1974*; *Meisel et al., 2005*). For neuroendocrine-mediated defects in the immune response the mechanisms that increase the occurrence of infections have been validated in animal models (*Prass et al., 2003*). It has been shown that immunosuppression in stroke manifests itself as lymphopenia and results in the conversion of lymphocyte Th1 phenotype to Th2 phenotype. Stroke has a severe effect on the normal balance between the nervous and immune systems, and immunodeficiency after stroke increases susceptibility to infection (*Dirnagl et al., 2007*). Immunosuppression and dysphagia were clearly associated independently with the risk of developing SAP in clinical studies (*Hoffmann et al., 2017*).

Stroke patients are often at risk of malnutrition, which is associated with disability in stroke patients, and 15% of them may have long-term dysphagia due to different stroke types and individual characteristics (*Beavan, 2015*; *Rowat, 2015*). Although nutritional screening methods are different, there is a high incidence of malnutrition in IS patients (*Yuan et al., 2021*; *Zhang et al., 2021*). Patients with swallowing dysfunction are at increased risk of developing pulmonary infections after stroke; however, even patients fed *via* total enteral nutrition are at risk of developing SAP, with nasogastric tubes promoting bacterial colonization of the oropharynx and predisposing patients to gastroesophageal reflux and vomiting, which can cause respiratory infections (*Brogan et al., 2014*; *Warusevitane et al., 2015*). Poor nutritional status is associated with the risk of complications related to poor stroke patient regression, including pneumonia.

PNI is calculated using two common laboratory indicators, ALB and LYM, which can be used to respond to a patient's immunonutritional status. Serum ALB plays neuroprotective roles in stroke, such as defending against oxidants and lowering hematocrit levels (*Belayev et al., 2001*). *Dziedzic (2004)* showed that ALB levels were significantly lower in acute stroke patients with poor prognosis and that low ALB levels increased the risk of disease. Reduced ALB levels are associated with worse nutritional status, and studies have shown that ALB can be used for nutritional assessment after acute stroke and has predictive value (*Gariballa et al., 1998*). In addition, low ALB levels are associated with decreased immunity, which can further increase the risk of pneumonia. Significant differences in prognosis of stroke patients by nutritional level at 7 d and 3 months (*Gariballa et al., 1998*). LYMs constitute a major class of immune cells involved in neuroinflammation and brain damage after stroke. Lymphopenia after stroke is common and is correlated with stroke severity. Researchers found that stroke patients had reduced peripheral LYMs in early studies of their immune status (*Członkowska, Cyrta & Korlak, 1979*; *Członkowska, Korlak & Kuczyńska, 1988*). *Kim et al. (2012)* reported that lower LYM in patients with AIS was associated with poorer functional prognosis. Considering the immunosuppression as well as the nutritional risk of SAP patients, the PNI may be a simple and practical objective indicator for assessing the prognosis of SAP patients. There was a significant reduction in PNI among SAP patients who died in the hospital as a result of our findings.

In the multivariate regression analysis, BMI was considered as an independent protective factor for in-hospital adverse events in SAP patients. The study found an increased risk of death in ischemic stroke patients with weight loss after 16 months of follow-up (*Wohlfahrt et al., 2015*). The meta-analysis reached similar conclusions, with stroke patients with reduced body weight having a significantly increased risk of death, and obesity playing a protective role in the prognosis of stroke patients (*Huang et al., 2016*). The prognosis of overweight or obese patients may be related to their greater metabolic reserve, which helps them cope with the metabolic imbalance caused by the disease (*Barba et al., 2015*; *Lavie, De Schutter & Milani, 2015*). We also found that hyperglycemia was an influential factor in the short-term poor prognosis of SAP patients, whereas the predictive value of T2DM was not found. As an acute cerebrovascular disease, elevated blood glucose in stroke patients can be attributed to poor treatment adherence leading to poor daily control and is also associated with stress hyperglycemia (*Dungan, Braithwaite & Preiser, 2009*; *QiMuge et al., 2022*). Excessive elevation of blood glucose promotes an inflammatory response as well as insulin resistance, while disrupting the immune system causing alterations in immune status, ultimately increasing the risk of pulmonary infections in patients. Similar to previous findings, increased blood glucose levels act as an independent risk factor promoting the development and progression of SAP in stroke patients (*Tshituta et al., 2019*; *You et al., 2019*).

## Limitations

It is important to note that this study has several limitations. Because this study was conducted at a single center, it lacks external validation. Although we adjusted for some

confounders, the effects of some potential confounders remain. We also did not dynamically monitor changes in the serum ALB concentration or LYM counts during admission, and the extent to which the PNI reflects the risk of mortality in SAP patients needs to be clarified in further studies. A multicenter study with a larger sample should confirm the relationship between nutritional status and in-hospital mortality in SAP patients.

## CONCLUSION

Malnutrition may promote the development of SAP, which in turn worsens the prognosis of stroke patients. Nutritional status is associated with a variety of factors, including inflammation, immunity, and metabolism, and patients with SAP are affected by multiple stroke comorbidities and have a greater need for nutritional status assessment. PNI, as a nutritional indicator calculated by a simple laboratory test, can be used for the assessment of short-term prognosis in patients with SAP. Timely interventions targeting high-risk patients may reduce the incidence of SAP and ultimately improve patient regression.

### Funding
The authors received no funding for this work.

### Competing Interests
The authors declare that they have no competing interests.

### Author Contributions
- Ke Xie conceived and designed the experiments, performed the experiments, analyzed the data, authored or reviewed drafts of the article, and approved the final draft.
- Chuan Zhang analyzed the data, prepared figures and/or tables, and approved the final draft.
- Shiyu Nie analyzed the data, prepared figures and/or tables, and approved the final draft.
- Shengnan Kang analyzed the data, prepared figures and/or tables, and approved the final draft.
- Zhong Wang analyzed the data, prepared figures and/or tables, and approved the final draft.
- Xuehe Zhang conceived and designed the experiments, performed the experiments, analyzed the data, authored or reviewed drafts of the article, and approved the final draft.

### Human Ethics
The following information was supplied relating to ethical approvals (*i.e.*, approving body and any reference numbers):

The Medical Ethics Review Committee of the Third People's Hospital of Chengdu (2024-S-245).

## Data Availability

The raw data are presented in Supplemental File 1.

## Supplemental Information

Supplemental information for this article can be found online at http://dx.doi.org/10.7717/peerj.19028#supplemental-information.

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
