# Peer review of "Prognostic nutritional index (PNI) as an influencing factor for in-hospital mortality in patients with stroke-associated pneumonia: a retrospective study"

_PeerJ, doi:10.7717/peerj.19028_

## Round 0.1 · original submission · Major Revisions

Thank you for submitting your manuscript. Please take careful note of the reviewers comments and recommendations should you revise your manuscript for further considerations.

Reviewer 1 ·

Basic reporting

The aim of this study was to evaluate the relationship between PNI levels and in-hospital mortality in SAP patients.
Major comments:
1. What has previously been published on this topic and what does this work add to the existing literature?
Liu, Y., Yang, X., Kadasah, S., & Peng, C. (2022). [Retracted] Clinical Value of the Prognostic Nutrition Index in the Assessment of Prognosis in Critically Ill Patients with Stroke: A Retrospective Analysis. Computational and Mathematical Methods in Medicine, 2022(1), 4889920.
Xiang, W., Chen, X., Ye, W., Li, J., Zhang, X., & Xie, D. (2020). Prognostic nutritional index for predicting 3-month outcomes in ischemic stroke patients undergoing thrombolysis. Frontiers in Neurology, 11, 599.
Liu, M., Sun, M., Zhang, T., Li, P., Liu, J., Liu, Y., ... & Ma, Y. (2023). Prognostic Nutritional Index (PNI) as a potential predictor and intervention target for perioperative ischemic stroke: a retrospective cohort study. BMC anesthesiology, 23(1), 268.
2. GBD showed that The annual number of strokes and deaths due to stroke increased substantially from 1990 to 2019(The Lancet Neurology, 20(10), 795-820). Data on the burden of stroke in your county should be increased so that readers can understand the need for stroke-related studies. Additional citations could be added: Estimated Burden of Stroke in China in 2020. JAMA Netw Open. 2023;6(3): e231455. doi:10.1001/jamanetworkopen.2023.1455’ and “China stroke surveillance report 2021. Military Med Res 10, 33 (2023). https://doi.org/10.1186/s40779-023-00463-x””
3. This study had been approved by the Ethics Committee? The approval no. of the Ethics Committee should be listed. Informed consent?
4. Is this a prospective or retrospective study? A flow chart may be more appealing to the reader. How many patients were enrolled and excluded? What were the reasons? Were any lost during follow-up and why?
5.When and how the blood samples were collected? Only one-time point was collected? How do biomarkers levels change over the course of follow-up?
6. How did treatment affect PNI? what was the effect of IV-tPA or endovascular thrombectomy on PNI? Did patients who were successfully reperfused after thrombectomy have lower/higher levels than those who were not?
7. This study showed that “PNI levels in SAP patients were associated with the risk of in-hospital mortality in patients, and increased PNI levels exerted an independent protective effect on the short-term prognosis of patients.” PNI levels were associated with the risk of in-hospital mortality in STROKE patients?
8. WHY PNI levels in SAP patients were associated with the risk of in-hospital mortality in patients? This reviewer would expect to see some points regarding how to translate these observations to help address this public health concern.

Experimental design

Original primary research within Aims and Scope of the journal.

Validity of the findings

Validity of the findings

Additional comments

no further comments

Reviewer 2 ·

Basic reporting

no comment

Experimental design

no comment

Validity of the findings

no comment

Additional comments

1. It is recommended to add a calibration curve to the established prediction model to evaluate the robustness of the model to further prove the predictive value of PNI for SAP death.
2. It is recommended to add experimental verification of the linear correlation analysis between PNI and SAP death.
3. The sample size is small, which may lead to serious data bias.
4.The method section does not mention the specific number of patients included in the analysis.
5. The description of the research subjects is a bit vague. What is the standard for judging lower respiratory tract infection?
6.BMI is related to the independent variable PNI of the study, and is also related to the dependent variable in-hospital death. It is a confounding factor. However, in the analysis, the author did not explain how to control this confounding factor. Please add it.
7.A total of 336 SAP patients were collected in this study, of which 30 died. Are the AUC values, sensitivity, and specificity of the prediction model the validation set? If so, how many deaths were there in the validation set?
8.Please add the ROC curve diagram

·

Basic reporting

1. In this paper, the authors summarize the hazards of SAP and explore the relationship between the PNI index and in-hospital mortality of SAP patients. Through a retrospective study, they conclude that elevated PNI levels are an independent protective factor for the risk of in-hospital death in SAP patients, and that the predictive value of PNI levels for in-hospital mortality in SAP patients is superior to that of single indicators, exhibiting high sensitivity and/or specificity. The research findings have certain clinical application potential, but there are still the following detailed issues.
2. The introduction mentions the concept of AIS, yet subsequent references remain centered on stroke. Since the study population mainly consists of AIS patients, it is recommended to add the differences in SAP incidence between hemorrhagic and ischemic strokes as a transition, or to replace "Stroke" with AIS later on.

Experimental design

3. The paper states that PNI mainly has clinical predictive value in cardiovascular, respiratory, and cancer patients, but does not provide research findings in AIS. It is recommended to include such conclusions and citations, or to establish a healthy control group for analysis in this study.
4. Whether the admission and exclusion criteria should exclude underlying respiratory diseases such as COPD needs consideration. It is advisable to refer to similar articles.
5. Regarding data collection, considering that PNI is a composite index of albumin (ALB) and lymphocytes (LYM), the analysis has not included data on nutritional methods and nutrient supplementation. Additional data that may contribute to SAP, such as the presence of gastric tubes, oral care, consciousness levels, and swallowing function, should also be included.
6. As in-hospital mortality due to SAP is the primary study outcome, it may be influenced by other comorbidities. It is suggested to include a specific analysis of such clinical data.

Validity of the findings

7. The paper still remains at a phenomenological level; it is recommended to further propose hypotheses regarding the mechanisms involved.
8. Some citations need updating, and the citation format is inconsistent.

·

Basic reporting

Article is well written. Authors may add how the findings of the study with help physicians in routine practice.

Experimental design

Kindly mention if patient of chronic liver disease and chronic kidney disease were excluded.

Addicting ROC curves may be better.

Validity of the findings

No comments

Additional comments

Article is good

---

## Round 0.2 · accepted · Accept

It appears that you have addressed the concerns of reviewers. Thank you for your persistence and patience during the submission process.

Reviewer 1 ·

Basic reporting

no comment

Experimental design

No comment'

Validity of the findings

I have nothing to add.

Additional comments

I have nothing to add.